# Intersection passing strategies for human-driven and autonomous vehicles in mixed traffic using DEA

**Jiajun Shen¤, Zhipeng Zhou¤\*, Guanyu Fu¤, Yu Wang¤**

College of Civil Science and Engineering, Yangzhou University, Yangzhou, China

¤ Current Address: College of Civil Science and Engineering, Yangzhou University, 196W. Huayang Road, Yangzhou, China.
\* zzpash@outlook.com

## Abstract

In this paper, we propose a right-of-way optimization model considering multi-objective DEA evaluation for intersections in mixed driving environments with automated and human driving. Considering average speed, number of cars, penetration of automated vehicles, queuing pattern, left-turn rate, and number of buses as factors influencing intersection rights-of-way. Comprehensively consider the per capita delay, travel time and traffic volume as the optimization objectives, and then determine the weights of the three optimization objectives for each strand of traffic flow, and calculate the cross-benefit by interchanging the weight evaluation through the Crossing Efficiency Evaluation Method (CREE) to determine the optimal order of traffic flow in each direction at the intersection. In this paper, the optimization strategy is compared with existing benchmarks (e.g., actuated control) using SUMO simulation software, and the simulation results show that the proposed optimization strategy is able to shorten the per capita delay and travel time at intersections in order to improve the efficiency of the traffic flow compared to actuated control and the First-Come, First-Served strategy.

## 1 Introduction

Nowadays, Connected Autonomous Vehicles (CAVs) are being phased into the market and are considered by many researchers as potentially bringing unprecedented changes to the transportation sector. Since CAVs reduce driver's inputs [1], they can improve safety by reducing human driver's errors to improve safety [2] and provide environmental benefits by reducing emissions and fuel consumption [3, 4].

The many benefits of CAVs need to be done with full market penetration, however, it is predicted that Human-driven Vehicles (HVs) will not be completely replaced by CAVs in the near future [5]. During the transition period, a mixture of human-driven and automated driving will be the most dominant form of road transportation.

**Data availability statement:** All relevant data are within the manuscript.

**Funding:** This research was funded by the Humanities and Social Sciences Research Planning Fund from the Ministry of Education of China, Grant No. 23YJAZH122. It was conducted as part of the project titled "Safety Assessment and Efficiency Optimization of Road Intersections in Mixed Traffic Environments of Manual and Autonomous Driving." The funders had no role in study design, data collection and analysis, decision to publish, or preparation of the manuscript.

**Competing interests:** The authors have declared that no competing interests exist.

Therefore, the interaction between CAVs and HVs is inevitable, and the uncertainty of driving behavior of HVs will severely limit the benefits of CAVs in improving traffic efficiency because HVs do not have a network connection [6, 7]. For example, at urban intersections, HVs' erratic speeds and lane-changing behaviors will prevent CAVs from forming stable convoys, resulting in the convoys not being able to keep moving efficiently and decreasing the efficiency of intersections. As a node connecting various roads, the intersection is a key component of the urban road network, and its traffic efficiency has a significant impact on the operation of the entire road network, making it very easy to become a bottleneck area of the urban road network. Therefore, it is important to develop intersection passing strategies that are adapted to mixed traffic flows.

The traditional method of determining the order of passage has obvious deficiencies in terms of comprehensiveness and is difficult to cope with the complexity of mixed traffic environments. Conventional methods tend to consider only a single or a few factors, ignoring the complexity and diversity of the transportation system, resulting in limited optimization. For example, the actuated control relies heavily on the traffic volume and velocity detected by the detectors to adjust the signal duration to determine the right-of-way for each traffic stream, and is unable to fully optimize the overall operational status of the intersection. The operational efficiency of intersections can be characterized by various output and input indicators. Data Envelopment Analysis (DEA) is a method that evaluates the decision effectiveness and relative efficiency of similar Decision Making Units (DMUs) by considering multiple indicators, making it well-suited for multi-objective optimization. Due to its capability to handle multi-input, multi-output problems, DEA is widely used in various applications. In traffic flow optimization, DEA serves as an effective multi-objective decision-making tool. By treating different lanes and signal cycles of an intersection as "decision units" and evaluating each intersection's efficiency based on indicators such as average delay, waiting time, and traffic volume, DEA can help identify the optimal traffic sequencing under multiple constraints.

The paper aims to research the passing strategies for urban intersections in mixed traffic environments in order to improve the intersection passing efficiency. Specifically, self-evaluation and cross-benefit evaluation of the weighting of indices through data envelopment analysis methods, combining self-assessment and other assessment to determine a relatively objective order of passage.

The innovation of this paper mainly lies in the use of data envelopment analysis, selecting three indexes of per capita delay, travel time and traffic volume, and obtaining the optimal traffic flow sequence of each approach at the intersection based on each index. The right-of-way optimization model considering multi-objective DEA evaluation considers the benefits of the three indices, assigns values to the optimal passing order under each index, and determines the passing order of traffic flows in each direction at the intersection. It is of great significance for the improvement of passing efficiency at intersections under mixed traffic environment of automated and human-driven vehicles.

The rest of the paper is organized as follows: Chapter 2 reviews the related studies on the passing strategies for automated vehicles at intersections. Chapter 3 presents the proposed DEA-based passing strategy. Chapter 4 conducts simulation experiments and results analysis using SUMO. Chapter 5 summarizes the research results of this paper.

## 2 Literature review

Various methods have been used to optimize the management and control of urban intersections, and in order to improve the operational efficiency of intersections, some researchers have modeled the trajectories of vehicles through intersections and solved optimization problems with safety constraints based on optimization methods. Yao and Li proposed a decentralized control model to optimize the trajectory of a CAV with the objective of minimizing the travel time, fuel consumption and safety risk of the CAV, and the results of the study showed that the decentralized control model outperforms the centralized control model in terms of both the computational efficiency and the benefits of the scheme compared to the centralized control model [8]. Chen et al. optimized the trajectory of the whole convoy by optimizing the trajectory of CAVs used to guide the speed based on the notion of "1+n" mixed platoon and designed a hierarchical event-triggered algorithm to optimize the signal control at intersections. The proposed control strategy outperforms the conventional control strategy in terms of average vehicle delay and fuel consumption [9]. Ma et al. developed a two-layer optimization model based on discrete time to plan the longitudinal and lateral trajectories of CAVs individually based on the distance between CAVs and stopping lanes, and their optimization strategy can significantly reduce the average delay of CAVs. And a mixed-integer linear programming (MILP) model with discrete time is proposed to jointly optimize automated intersection management and traffic signal design, aiming to simultaneously improve traffic efficiency, fuel economy, and driving comfort [10]. Wu and Jiang proposed a mixed-integer linear programming model with discrete time to co-optimize automated intersection management and trajectory smoothing designs aiming to simultaneously improve traffic efficiency, fuel economy, and driving comfort [11]. Jiang et al. proposed a two-layer cooperative control strategy that prioritizes any two conflicting motions according to lateral and tailgating safety constraints in the upper layer, and optimizes the speed profiles of the convoy-leading vehicles controlled by the upper layer's dynamic prioritization in the lower layer to minimize the energy consumption, and the proposed approach significantly reduces the total delays as compared to the existing strategies with the first-come-first-served policy [12]. Li et al. used an optimization method based on genetic algorithm to determine the vehicle passing order while calculating the optimal vehicle trajectory, and the results of the study showed that the proposed algorithm was able to reduce the average travel time delay at intersections by 16.3% to 79.3% compared to inductive signal control [13]. Feng et al. estimated the arrival time and departure time of each vehicle at an intersection based on the trajectory data of the last stopped human-driven vehicle and the first unstopped human-driven vehicle, thus optimizing the signal timing to minimize the total vehicle delay [14]. Feng et al. proposed an adaptive coupling control method based on vehicle queuing to optimize signal timing and vehicle trajectories in mixed traffic, and the results showed that the method can significantly reduce delays and fuel consumption under high CAV penetration and high traffic volume [15]. Kamal et al. Used a two-way communication network to solve the optimal trajectory of vehicles through an intersection based on safety constraints in a model predictive control framework to achieve global coordination of vehicles in all directions within the intersection, which avoided collisions while also improving the efficiency of vehicles passing through the intersection, and the proposed scheme significantly improved the performance of the intersection as compared to the conventional signalized intersection scheme [16]. Lu et al. proposed an optimization-based method to provide the fastest discrete-time trajectories for vehicles wanting to pass through an intersection, and the results showed that the overall traffic control performance of this optimization method was significantly better than that of the conventional traffic signal control method [17].

A large number of researchers have based the reservation-based approach on the First-Come-First-Served (FCFS) intersection control strategy, which organizes the vehicles to occupy the intersection space sequentially to coordinate the vehicles to pass through the intersection without conflicts. Dresner et al. proposed a model that is widely used in the field

of autonomous driving. The core of this model is to grid the intersection area, and according to the sequence of vehicles arriving at the grid, using the FCFS control policy for the allocation of spatial and temporal resources, which rules that the automated vehicles must submit a request for passage to the traffic management center in a certain time sequence, and the vehicle can enter the intersection and continue to drive only after the control center approves the vehicle's request for passage. Experiments demonstrate that the FCFS strategy can reduce delays in most cases [18]. Huang et al. building on the research by Dresner et al., proposed assigning different priorities to vehicles approaching intersections by providing a speed range for the vehicles and implementing a dynamic reservation protocol. The results demonstrated that this control method offers significant environmental benefits [19]. Levin and Rey used a mixed-integer linear programming model to assign intersection reservation priorities to CAVs, and showed that the model provided an optimization strategy that significantly reduced intersection delays [20]. Bisht and Shet prioritized the convoys based on their proximity to the intersection, and the simulation results show that the strategy can effectively improve the intersection capacity [21].

A substantial body of research is based on planning methods, where the sequence of vehicles passing through intersections is scheduled according to different objectives. Zhou et al. proposed an unsignalized intersection control framework consisting of microscopic virtual fleet control and macroscopic traffic flow regulation to determine the fleet's passing strategy with the objective of optimal traffic volume, and the results showed that the framework can effectively mitigate intersection delays through traffic flow regulation [22]. Chen et al. using graph analysis, modeled vehicle trajectory conflicts as a conflict directed graph and a coexistence undirected graph. They proposed an improved depth-first search spanning tree algorithm and a minimum clique cover algorithm to determine the optimal passing sequence. Simulations were conducted for varying numbers of vehicles and traffic volumes, and the results demonstrated that the passing strategies provided by this model significantly reduce vehicle delays at intersections [23]. Xu et al. proposed an unsignalized intersection cooperative driving strategy based on a tree table in the passing order solution space, combined with Monte Carlo tree search and some heuristic rules, and the proposed strategy is able to find a nearly globally optimal passing order in a very short planning time [24]. Hu et al. proposed a multi-vehicle collaboration method for signal-free intersections based on the concept of virtual fleet, which uses the Constraint-Tree-driven Modeling (CTM) strategy to determine the conflict-free passing order and driving mode of vehicles at the supervisory level, and the Distributed Robust Control (ORC) method to control the vehicles through the intersection at the execution level. Distributed Robust Control (ORC) method is utilized in the execution layer to control the vehicles passing through the intersection, and the results show that the proposed method significantly improves the fuel economy, driving comfort and safety under low traffic volume [25].

Recent studies have extensively explored optimization strategies in mixed environments of autonomous and human-driven vehicles. Wan et al. proposed a multi-objective optimization model for mixed traffic, with average travel time and fuel consumption as optimization objectives, which significantly improved traffic efficiency [26]. Wang et al. introduced a traffic flow adjustment multi-objective optimization model for mixed environments, employing a bi-level programming approach and using subsidy nodes to regulate traffic flow, thereby optimizing network performance, emissions, and equity [27]. Moon et al. proposed a multi-objective vehicle path control method for regulating intersection travel directions in mixed traffic environments based on reinforcement learning, designing a reward function with travel speed and distance to destination as optimization objectives. Simulation results demonstrated that the proposed method outperformed existing approaches in terms of autonomous vehicle travel distance, time, and waiting time, particularly in more dynamic traffic environments [28].

Most of the existing researches on intersection passing strategies focus on automated vehicles, and determine the passing sequence of automated vehicles through centralized or distributed control, aiming to improve the operational efficiency of automated vehicles themselves. In addition, although existing multi-objective optimization studies provide valuable models and strategies for mixed traffic environments, most of these studies focus on specific optimization objectives, such as travel time and fuel consumption, without fully integrating optimization objectives that reflect intersection passage efficiency. Furthermore, some studies rely on methods such as reinforcement learning, which face challenges

related to large-scale data training and model accuracy. The complex traffic environment at intersections is difficult to accurately map to the states, actions, and rewards of reinforcement learning agents, potentially resulting in suboptimal training outcomes. The contribution of this study lies in the proposed optimization model, which comprehensively considers various factors, including average speed, number of cars, penetration of automated vehicles, queuing pattern, left-turn rate, and number of buses. This model provides a more holistic representation of the complex traffic flow characteristics at intersections. Additionally, the optimization objectives focus on improving intersection passage efficiency, incorporating multi-dimensional goals such as per capita delay, travel time, and traffic volume, thereby making the optimization results more comprehensive and accurate. These influencing factors and optimization objectives provide a more precise and scientifically grounded basis for optimizing intersection passage efficiency. The proposed method allows for the rapid and accurate determination of the optimal passage order for each direction at the intersection, without the need for extensive data training.

## 3 Methodology

In this study, left-turning and through vehicles on the same approach are released simultaneously to avoid conflicts. This study assumes that all vehicles travel along a single lane, which simplifies the two-dimensional motion of vehicles within the intersection. Although this assumption may not fully capture the complexity of real-world scenarios, existing studies have demonstrated that lane-based models effectively capture the key dynamic characteristics of intersection traffic flow [29, 30]. Future research could further extend the model to account for the two-dimensional movement of vehicles within the intersection. When a platoon reaches the intersection, it is essential to determine the order of passage for arriving platoons from each approach to ensure efficient traffic flow. The intersection's roadside unit is responsible for gathering platoon information from each approach, analyzing and processing this data to determine the optimal order of passage. The sequence of traffic signal phases is then adjusted to control the order of vehicle passage. Vehicles with a headway of less than 3 secs are grouped into a single platoon. Each signal cycle allows one platoon to pass through each approach, and the passing order is recalculated for each subsequent cycle. In this study, average speed, number of cars, penetration of automated vehicles, queuing pattern, left-turn rate, and number of buses are selected as the influencing factors for researching the development of intersection passing rules, and using Data Envelopment Analysis (DEA) to comprehensively consider a variety of traffic flow optimization indices, such as per capita delays, passing time and traffic volume, etc., and ultimately optimize by determining the weights of the indices and adjusting the weights of the different objectives the traffic sequence of each direction of traffic flow at the intersection.

This section details the methodology used to determine intersection passing strategies in mixed environments.

### 3.1 Influencing factors

Formulating traffic strategies requires comprehensive consideration of multiple factors.

Average speed: Different average speed directly affects the capacity and efficiency of intersections, priority should be given to vehicles with higher speeds, as this can effectively reduce vehicle delays at the intersection and improve overall throughput efficiency.

The number of cars and buses: Excessive intersection flow can lead to congestion and increased vehicle delays. Prioritizing the release of traffic from the approaches with higher traffic volumes can help reduce delays at the intersection. Besides, due to the large passenger volume of buses, priority at intersections can effectively reduce per capita delays and improve passage efficiency.

Queuing patterns and the penetration of automated vehicles: At intersections where automated and human-driven vehicles are mixed, their queuing patterns and the penetration of automated vehicles have different impacts on intersection efficiency due to the different driving characteristics of the two types of vehicles. The queuing pattern is the specific queuing order of the two types of vehicles in a mixed traffic flow containing CAVs and HVs. The queueing patterns can be

categorized into three types: the first type, where all CAVs are positioned at the front of the queue; the second type, where CAVs and HVs are randomly mixed within the queue; and the third type, where all HVs are positioned at the front of the queue.

Left-turn rate: Turning vehicles need to wait for the right time to complete the turn and their speed will be reduced, this process will cause other vehicles to slow down or stop and wait, thus prolonging the passage time and reducing the efficiency of the intersection.

In summary, in this research, average speed, number of cars, penetration of automated vehicles, queuing pattern, left-turn rate, and number of buses were selected as influencing factors for the development of intersection passing strategies.

## 3.2 Optimization objectives

The most common method used in determining the order of passing is based on the arrival time of vehicles at the intersection, i.e., First-Come-First-Served (FCFS) strategies. Although the method is simple and easy to apply, it is difficult to maximize the efficiency of the intersection in some cases. In order to realize the optimal passage effect of the intersection, it is necessary to consider it comprehensively from several perspectives.

From the perspective of road users, it is crucial to reduce per capita delays, as this metric serves as an important indicator of overall intersection operational efficiency, particularly in scenarios involving buses. High per capita delays typically signify low traffic efficiency, while prioritizing high-capacity buses can significantly reduce per capita delays, thereby improving intersection service levels. The per capita delays metric ensures the relative priority of public transportation in mixed traffic conditions when determining the passage order, thus contributing to enhanced overall traffic flow efficiency.

From the perspective of maximizing the intersection's overall benefit, it is necessary to minimize vehicle travel time and maximize traffic throughput. While buses can reduce per capita delays, their slower speeds and frequent stops often result in longer travel times. Solely considering delay time neglects the impact of travel speed. Therefore, incorporating travel time into the optimization process provides a more comprehensive reflection of the timeliness of traffic flow. Traffic volume is a core indicator of intersection capacity, directly influencing saturation and congestion levels. In a mixed-traffic environment, high traffic volumes may lead to severe congestion, affecting the performance of prioritized vehicles. By accounting for traffic volume in optimizing the passage order, the flow demands of different traffic types can be balanced, enabling reasonable flow distribution and ensuring the maximization of overall intersection throughput.

Table 1 summarizes the main parameters used in the optimization objectives calculation.

**3.2.1 Per capita delays.** Due to the high passenger capacity of buses and their ability to move large numbers of passengers in a short period of time, per capita delay rather than average vehicle delay was used as the assessment index. Higher bus volume percentages can be effective in reducing per capita delays, and priority releases can be considered if an intersection has a higher bus percentage of traffic flow in a particular direction. Formula for calculating delays per capita:

$$\bar{d} = \frac{d_g}{p_g} = \frac{\sum_i d_i \times p_i}{\sum_i p_i}$$

(3-1)

where $p_i$ is a known quantity that can be acquired in real time by the in-vehicle detector. When vehicles pass through intersections, travel time delays occur due to acceleration and deceleration, stopping, and queuing. Formula for calculating the single-vehicle delay for the $i$th vehicle:

$$d_i = T - T'$$

(3-2)

The state of vehicles arriving at the intersection varies, necessitating the calculation of individual vehicle delay $d_i$ under different conditions.

**Table 1. Parameters for optimization objectives calculation.**

| Parameters | Description |
|---|---|
| $\bar{d}$ | Per capita delays (s) |
| $dg$ | Total traffic delays (s) |
| $pg$ | Total number of persons in the traffic flow (person) |
| $di$ | Single-vehicle delay for vehicle $i$ (s) |
| $pi$ | Number of passengers in vehicle $i$ (person) |
| $T$ | Actual travel time of vehicles through the intersection (s) |
| $T'$ | Vehicle travel time through intersections operating in normal conditions (s) |
| $T'_i$ | Travel time for the vehicle $i$ to travel through the intersection in normal conditions of operation (s) |
| $T_i$ | Travel time for the vehicle $i$ to travel through the intersection in real conditions of operation (s) |
| $l_i$ | Distance traveled by vehicle $i$ adjusting its normal speed to the speed at which it passes through the intersection (m) |
| $s_i$ | Distance traveled by the vehicle $i$ through the intersection at the recommended speed u (m) |
| $m_i$ | Distance traveled by vehicle $i$ adjusted from the speed at which it passed through the intersection to its normal travel speed (m) |
| $v_i$ | Normal speed of vehicle $i$ (km·h$^{-1}$) |
| $u_i$ | Recommended speed of the vehicle $i$ passing through the intersection (km·h$^{-1}$) |
| $ad$ | Deceleration (m·s$^{-2}$) |
| $au$ | Acceleration (m·s$^{-2}$) |
| $t_{iw}$ | Waiting time for the vehicle $i$ in the waiting line for stopping (s) |
| $t_l$ | The average travel time for a vehicle to cover the distance $l$ from the point of detection to the stop line (s) |
| $l$ | The distance from the location where the roadside unit at the intersection detects a vehicle to the stop line (m) |
| $v$ | Average free travel speed of vehicles at an approach at intersections (m·s$^{-1}$) |
| $t'_l$ | Average travel time for the distance traveled by a vehicle from the stop line to the point of conflict (s) |
| $e$ | Automated vehicles penetration rate (%) |
| $f$ | Indicates the queuing configuration, the queuing configuration is divided into three queuing modes under this index, the first one is all CAVs in the front, assigned a value of 1; the second one is a random mix of CAVs and HVs, assigned a value of 2; the third one is all HVs in the front, assigned a value of 3 |
| $g$ | Left-turn rate, as a percentage of the number of vehicles turning left at an approach to the total number of vehicles at the intersection (%) |
| $Ve$ | Passenger Car Unit (veh·pcu$^{-1}$) |
| $V$ | Total traffic volume (veh·pcu$^{-1}$) |
| $Pj$ | Percentage of category $j$ vehicle traffic to total traffic volume (%) |
| $Ej$ | Vehicle conversion factor for category $j$ |

Scenario 1: Straight through without stopping for queues, as shown in Fig 1.Travel time for the vehicle $i$ to travel through the intersection in normal conditions of operation:

$$T'_i = \frac{l_i + s_i + m_i}{v_i} \tag{3-3}$$

$$l_i = \frac{v_i^2 - u_i^2}{3.6^2 \times 2a_d} \tag{3-4}$$

$$m_i = \frac{u_i^2 - v_i^2}{3.6^2 \times 2a_u} \tag{3-5}$$

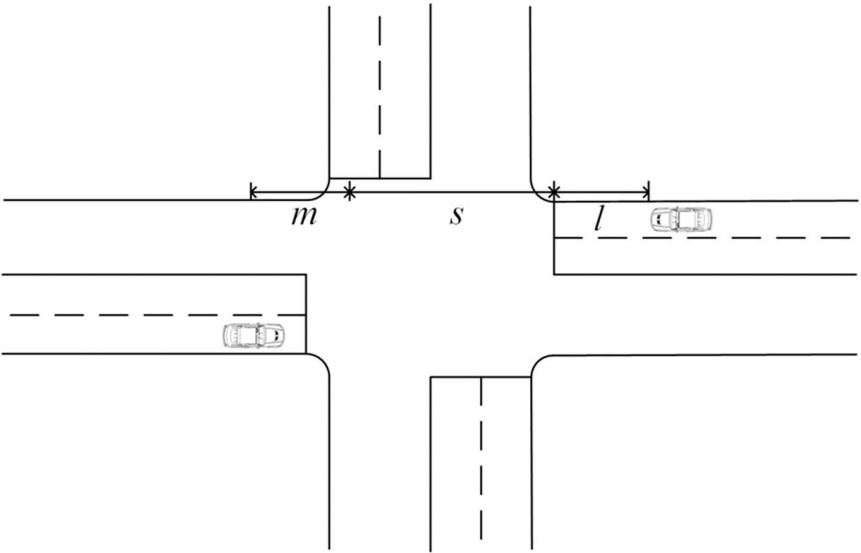

**Fig 1. Schematic of scenario 1 operation.**

Travel time for the vehicle $i$ to travel through the intersection in real conditions of operation:

$$T_i = \frac{v_i - u_i}{a_d} + \frac{s_i}{u_i} + \frac{v_i - u_i}{a_u}$$

(3-6)

Single-vehicle delay for vehicle $i$ :

$$d_i = \frac{s_i(u_i - v_i)}{u_i v_i} + \frac{u_i - v_i}{3.6^2 \times 2a_u} - \frac{v_i}{2a_d} - \frac{u_i^2}{3.6^2 \times 2a_u \times v_i}$$

(3-7)

Scenario 2: Passing after stopping and queuing, as shown in Fig 2.The time for a vehicle to accelerate to speed $u$ based on Scenario 1:

$$t_s' = \frac{u_i}{a_d}$$

(3-8)

Travel time for the vehicle $i$ to travel through the intersection in real conditions of operation (including the actual stopping and waiting time):

$$T_i = \frac{v_i}{a_d} + \frac{u_i}{a_d} + \frac{s_i - \frac{v_i^2}{3.6^2 \times 2a_d}}{u_i} + \frac{v_i - u_i}{a_u} + t_{iw}$$

(3-9)

Single-vehicle delay for vehicle $i$:

$$d_i = \frac{2v_i}{a_d} + \frac{s_i(v_i - u_i)}{u_i v_i} - \frac{v_i(v_i + u_i)}{3.6^2 \times 2a_d \times u_i} - \frac{v_i^2 - u_i^2}{3.6^2 \times 2a_u \times v_i} + t_i$$

(3-10)

Bring the parameters of each traffic flow into the above two scenarios and calculated to obtain the order of passing considering per capita delay.

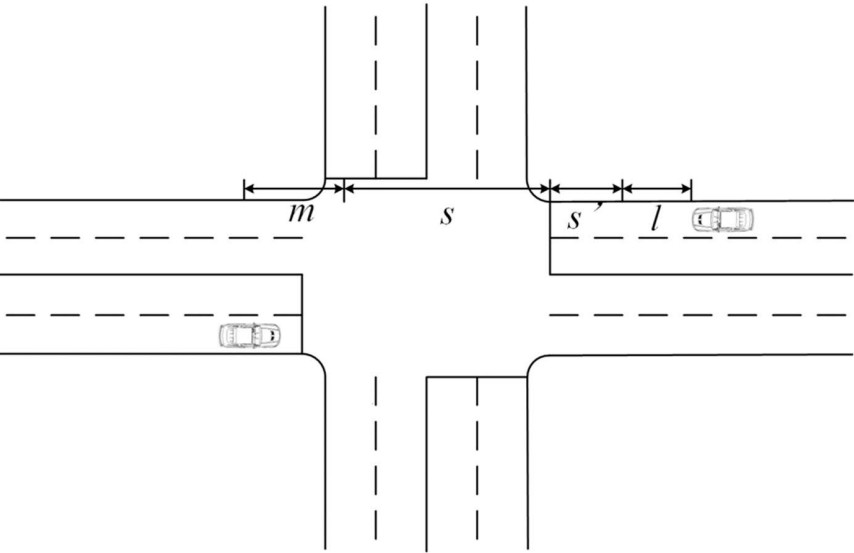

**Fig 2. Schematic of scenario 2 operation.**

**3.2.2 Travel time.** The high passenger capacity of buses can effectively reduce per capita delays, resulting in a relatively low overall delay level. However, buses often experience longer travel times due to frequent stops and slower speeds. Therefore, relying solely on either per capita delays or travel time as a single indicator cannot fully capture the intersection's operational efficiency. Combining both metrics provides a more accurate assessment of the impact of different vehicle types on the overall traffic flow. Calculating travel time can be done by using possible conflict points in the traffic flow to calculate the time it takes for a vehicle to pass through an intersection. A conflict point is a service window in a queueing system with multiple single-channel service windows.

The conflict point used to calculate elapsed time in this section is the last conflict point before the vehicle exits the intersection. Determining the passing sequence based on the FCFS principle can consider calculating the travel time of vehicles from the point of detection to passing the conflict point, in order to establish the anticipated arrival order of the two conflicting traffic streams at the conflict point, as shown in Fig 3.

The travel time $t_i$ for the vehicle $i$ is divided into the average travel time $t_l$ for covering the distance $l$ from detection to the stop line, and the average travel time $t'_l$ for covering the distance $l$ from the stop line to the conflict point:

$$t_i = t_l + t'_l \tag{3-11}$$

$l$, $l'$ are shown in Fig 3, and the travel time $t_l$ is expressed by the following equation:

$$t_l = \frac{3.6 \times l}{v} \tag{3-12}$$

Travel time $t_2$ is calculated by combining the effects of three parameters: automated vehicle penetration $e$, queuing configuration $f$, and left-turn rate $g$. With the increasing penetration of autonomous vehicles, the coordination among these vehicles is enhanced, leading to reduced reaction times and shorter travel times. Autonomous vehicles are able to maintain optimal speeds and intervals, minimizing delays caused by the unstable driving behaviors of vehicles. Consequently, as the penetration rate of autonomous vehicles rises, travel times will decrease. The queuing configuration is the specific queuing order of the two types of vehicles in a mixed traffic flow containing

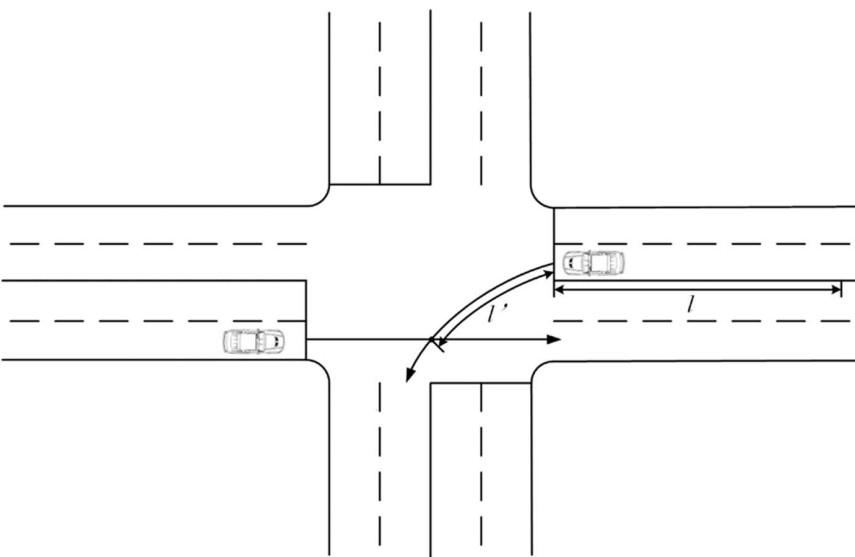

**Fig 3. Schematic of travel time calculation.**

CAVs and HVs. In this section the queuing configuration $f$ is divided into three types, the first with all CAVs in front and assigned the value 1, the second with a random mixture of CAVs and HVs and assigned the value 2, and the third with all HVs in front and assigned the value 3. In addition, the presence of turning vehicles in the flow of traffic is also an important factor affecting the sequence of passing. Turning vehicles need to wait for the right opportunity to complete the turn and their speed will be further reduced, which may lead to other vehicles slowing down or stopping to wait, thus prolonging the travel time. Therefore, the left turn rate $g$ also becomes one of the parameters considered, with a specific value of the number of left-turning vehicles in the traffic flow as a percentage of the total number of vehicles in the traffic flow.

Simulations were performed using SUMO simulation software to obtain different travel times by adjusting the rate of penetration $e$, the queuing configuration $f$ and the left-turn rate $g$, respectively. Subsequently, the penetration rate, queuing configuration, left-turn rate, and travel time were fitted, yielding a goodness-of-fit $R^2 = 0.912$. The resulting index function is given by:

$$t'_l = -0.0131e + 0.435f + 0.018g + 4.219$$

(3-13)

**3.2.3 Traffic volume.** Traffic flow is a key factor in determining the order of traffic flow at each approach of the intersection, the volume of which directly affects the capacity and efficiency of the intersection. Excessive traffic volumes can lead to severe traffic jams and significantly reduce the efficiency of travel. Therefore, determining the order of passage by comparing the traffic volume of each approach roadway can effectively improve the overall efficiency of the intersection.

The traffic flow in this research takes into account the presence of buses, so it is necessary to calculate the passenger car unit in each direction of the intersection by the vehicle conversion factor, which is shown in Table 2.

Passenger car unit is calculated as follows:

$$V_e = V \sum P_j E_j$$

(3-14)

**Table 2. Common vehicle conversion factors.**

| vehicle type | Vehicle conversion factor |
| --- | --- |
| car | 1.0 |
| bus | 1.5 |

### 3.3 The right-of-way optimization model considering multi-objective DEA evaluation

Data Envelopment Analysis (DEA) is a systematic analysis method based on the concept of "relative efficiency", which is used to evaluate the relative effectiveness or efficiency of units of the same type through multi-index inputs and multi-index outputs. DEA is especially suitable for complex systems with multiple inputs and multiple outputs, and it is mainly used for the relative ranking and benefit evaluation of Decision Making Unit (DMU). The classical DEA model is mainly a self-assessment work based on its own preference for each index, and its results tend to lack credibility due to high subjective factors, so this research adopts the Cross-Efficiency Evaluation (CREE) method [31], which is the work of cross-swapping the weights among DMUs that combines self-assessment and other assessment for a relatively objective ranking under recognized weights. To ensure clarity, all notations used in this section are firstly summarized in Table 3

**3.3.1 Model formulation.** The right-of-way optimization model considering multi-objective DEA evaluation can be written as:

$$\psi = G(P, U, W) \tag{3-15}$$

The set of programs $P$ is a set containing all the traffic flows at the intersection, where each program $p_i$ represents the traffic flow in a certain direction at the intersection. In this set, each flow is considered as a Decision Making Unit (DMU), i.e., $DMU_i = p_i$. The attribute index set $U$ contains all the indices used to evaluate and compare different programs. Each element $u_i$ represents the set of attribute indices of program $p_i$, i.e., $u_i = \{u_{i1}, u_{i2}, \ldots, u_{im}\}$。These indices reflect the performance of different programs. Among these indices, objectives that require maximization of values are used as output indices, while those that require minimization of values are used as input indices. In the model in this section, two input indices and one output index are set, i.e., delay per capita, travel time, and traffic flow, respectively. As shown in Table 4:

The value of the I/O index for each DMU$_i$ is determined by the relative priority achieved by each approach traffic flow in each index computation, e.g., in an input indicative index p computation, $p_1 > p_2 > p_3 > \cdots > p_n$, there is an assignment

**Table 3. Notations.**

| Notations | Description |
| --- | --- |
| P | $P = \{p_1, p_2, \ldots, p_n\}(n \geq 2)$ is the set of programs |
| U | $U = \{u_1, u_2, \ldots, u_n\}(n \geq 2)$ is the set of attribute indices |
| W | $W = \{w_1, w_2, \ldots, w_n\}(n \geq 2)$ is the set of weights |
| $\theta i$ | Self-assessed benefit value of the $i$th decision-making unit |
| $I_{jp}$ | $p$th input index for the $j$th decision unit |
| $O_{jq}$ | $q$th output index for the $j$th decision unit |
| $a_{ip}$ | Weight of the $p$th input index for the $i$th decision unit |
| $\beta_{iq}$ | Weight of the $q$th input index for the $i$th decision unit |
| $v_{Ip}^{j}$ | Benefits of individual input attribute indices |
| $v_{Op}^{j}$ | Benefits of individual output attribute indices |
| $\Omega ij$ | The combined benefit of $p_j$ to $p_i$, i.e., the ratio of the combined output benefit $O_i^j$ to the combined input benefit $I_i^j$ |

**Table 4. I/O indices for each DMU.**

| DMU | input indices | | output indices |
|---|---|---|---|
| $p_1$ | $\bar{d}_1$ | $t_1$ | $V_{e1}$ |
| $p_2$ | $\bar{d}_2$ | $t_2$ | $V_{e2}$ |
| ... | ... | ... | ... |
| $p_n$ | $\bar{d}_n$ | $t_n$ | $V_{en}$ |

constraint $I_{1p} > I_{2p} > I_{3p} > \cdots > I_{np}$. The weight set W contains the weights of each input and output index, i.e., $w_i$ is weighted as $\alpha_{ip}$ for the input index $I_{ip}$ and $\beta_{iq}$ for the output index $O_{iq}$. The model for calculating the weights of each DMU index is expressed as follows:

$$\theta_i = \max \frac{\sum_{q=1}^{h} (\beta_{iq} O_{iq})}{\sum_{p=1}^{l} (\alpha_{ip} I_{jp})}$$

(3-16)

$$s.t. \begin{cases} \sum_{q=1}^{h} (\beta_{iq} O_{iq}) - \sum_{p=1}^{l} (\alpha_{ip} I_{jp}) \leq 0 \\ j = 1, 2, ..., n \\ \sum_{p=1}^{l} (\alpha_{iq} I_{jp}) = 1 \\ \beta_{iq}, \alpha_{ip} \geq 0 \\ p = 1, 2, ..., l \\ q = 1, 2, ..., h \end{cases}$$

(3-17)

### 3.3.2 Benefits evaluation.

(1) Calculation of benefit values

The set of weights derived from Equations 3–16 and 3–17 is subjective, so it is necessary to calculate the cross-benefits by evaluating the weights of DMUs interchangeably with each other through the CREE method, i.e., the weights of $\text{DMU}_j$ (j = 1,2,3...n) are used to evaluate the $\text{DMU}_i$.

The evaluation of the benefits of each traffic flow is "the weight of each objective × the index value of each objective", and the benefits of individual input or output attribute indices are shown in the following equations, respectively:

$$v_{Ip}^{j} = \alpha_{jp} \times I_{ip}$$

(3-18)

$$v_{Op}^{j} = \beta_{jq} \times O_{iq}$$

(3-19)

When i = j, it is the self-evaluation of each flow considering the weights of each I/O objective; when i ≠ j, it is the other-evaluation of each I/O objective of each flow on the weights of other flows.

(2) Determining the order of passage

The comprehensive benefit $\Omega_{i,j}$ of $p_j$ to $p_i$ is the ratio of the comprehensive output benefit $O_{i,j}$ to the comprehensive input benefit $I_{i,j}$, which is calculated as follows:

$$O_i^j = \sum_{q=1}^{1} (\beta_{jq} \times O_{iq})$$

(3-20)

$$I_i^j = \sum_{p=1}^{2} (\alpha_{jp} \times I_{ip})$$

(3-21)

$$\Omega_{i,j} = \sum_{q=1}^{1} (\beta_{jq} \times O_{iq}) / \sum_{p=2}^{2} (\alpha_{jp} \times I_{ip})$$

(3-22)

Under group decision making, there is both a comprehensive benefit self-assessment of all objectives by a traffic flow decision maker according to its own weight and mutual assessment of the same objective under the weights of different right-of-way decision makers, which forms the comprehensive benefit vector set$\Omega = (\Omega_1, \Omega_2, \ldots, \Omega_n)^T$ of this model, and the comprehensive benefit matrix is expressed as follows:

$$\Omega = \begin{bmatrix} \Omega_1^1 & \Omega_1^2 & \cdots & \Omega_1^j & \cdots & \Omega_1^n \\ \Omega_2^1 & \Omega_2^2 & \cdots & \Omega_2^j & \cdots & \Omega_2^n \\ \vdots & \vdots & \ddots & \vdots & \vdots & \vdots \\ \Omega_i^1 & \Omega_i^2 & \cdots & \Omega_i^j & \cdots & \Omega_i^n \\ \vdots & \vdots & \ddots & \vdots & \vdots & \vdots \\ \Omega_n^1 & \Omega_n^2 & \cdots & \Omega_n^j & \cdots & \Omega_n^n \end{bmatrix}$$

(3-23)

The order in which each flow passes is determined by calculating the magnitude of the average value $\bar{\Omega}_i$ of each item of each row vector.

$$\overline{\Omega_i} = \frac{1}{n} \sum_{j=1}^{n} \Omega_i^j$$

(3-24)

### 3.4 Model feasibility analysis

The right-of-way optimization model for each approach of the intersection was explained in the previous section, and the feasibility of this optimization model is examined in this section. This model selects 6 traffic flow parameters (average speed, queuing pattern, automated vehicle penetration rate, number of cars, number of buses, and left-turn rate) to calculate the value of I/O indices for each approach of the intersection, and when one of the parameters is changed and the rest are fixed, it is able to directly determine the order of the traffic flow at each approach of the intersection. By substituting the above parameters into the model, the optimal passing order can be calculated, and the feasibility of the model can be verified by comparing the calculated results with the directly judged passing order.

The intersections used for all the examples in this section are shown in Fig 4. From the west approach of the intersection, the approaches are numbered clockwise, i.e., the west approach is numbered 1; the north approach is numbered 2; the east approach is numbered 3; and the south approach is numbered 4. When the vehicle enters the approach lane, the speed of the vehicle is set within the speed range of 30 km/h to 50 km/h. In order to simulate the dynamic behavior of the vehicle, the acceleration and deceleration when passing through the intersection are uniformly set to 2 m/s². In addition, different types of vehicles were given different passenger capacities, set at 1per/veh for cars and 25per/veh for buses.

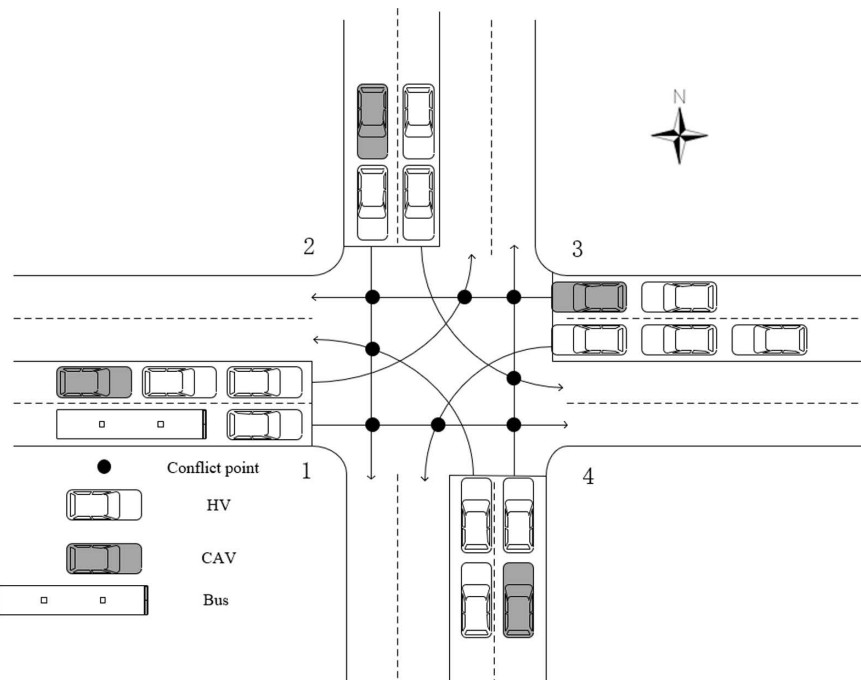

**Fig 4. Intersection Schematic.**

Example 1: Different number of buses

(1) Parameter calibration

1, 2, 3, and 4 buses are assigned in each direction of the intersection in sequential order, with the rest of the parameters being consistent. Traffic flow data for each approach of the intersection during a pass cycle, as shown in Table 5.

(2) Calculating the optimization objective value

Bring the above parameters into the calculation formula under each objective and the results are shown in Table 6.
As can be seen from Table 5, except for the values of the travel time indices, which are all equal, the order of travel determined by ranking the per capita delay indices from lowest to highest and the traffic volume indices from highest to lowest is the same, as follows: 4 → 3 → 2 → 1。

(3) Determining the optimal right-of-way

According to each of the three indices corresponds to the optimal order of passing to the intersection of each approach traffic volume from high to low, respectively, as the value of each direction of traffic flow I/ O indices, due to the intersection of each direction of the travel time index calculation results are equal, the value of its assigned to 1 to participate in

**Table 5. Parameters for each approach of the intersection.**

| | Average speed/km·h⁻¹ | Queuing pattern | Automated vehicle penetration rate/% | Number of cars/veh | Number of buses/veh | Left-turn rate/% |
|---|---|---|---|---|---|---|
| 1 | 40 | 2 | 0.5 | 8 | 1 | 50 |
| 2 | 40 | 2 | 0.5 | 8 | 2 | 50 |
| 3 | 40 | 2 | 0.5 | 8 | 3 | 50 |
| 4 | 40 | 2 | 0.5 | 8 | 4 | 50 |

**Table 6. Index value of each approach at intersections.**

| | Per capita delays (—d·s⁻¹) | Travel time (t·s⁻¹) | Traffic volume (veh.pcu⁻¹) |
|---|---|---|---|
| 1 | 1.08 | 14.33 | 9.5 |
| 2 | 0.99 | 14.33 | 11 |
| 3 | 0.93 | 14.33 | 12.5 |
| 4 | 0.88 | 14.33 | 14 |

**Table 7. Indices of traffic flow in each direction and the index weights.**

| | Per capita delays | Travel time | Traffic volume |
|---|---|---|---|
| 1 | 1(0.40) | 1(0.21) | 1(0.39) |
| 2 | 2(0.42) | 1(0.17) | 2(0.41) |
| 3 | 3(0.46) | 1(0.15) | 3(0.39) |
| 4 | 4(0.48) | 1(0.12) | 4(0.40) |

the subsequent determination of the weights. Through Equations 3–16 and 3–17, the weights of the I/O indices for each direction of traffic flow are obtained, and the results are shown in Table 7.

Based on the I/O indices and their weights, along with Equations 3–20, 3–21, 3–22, and 3–24, the cross-efficiency matrix is obtained. Calculate the mean value of the elements of each row of the matrix as the comprehensive efficiency of each approach lane of the intersection, and order the comprehensive efficiency from highest to lowest to obtain the order of traffic flow of each approach lane. The efficiency value of the west approach is calculated to be 0.69, the efficiency value of the north approach to be 0.76, the efficiency value of the east approach to be 0.79, and the efficiency value of the south approach to be 0.81, and thus the final order of passage is 4→3→2→1, i.e., south approach →east approach →north approach →west approach.

Example 2: Different automated vehicle rates of penetration

(1) Parameter calibration

The penetration rates were set to 0.125, 0.375, 0.625, and 0.875 in each direction of the intersection in sequential order, and the rest of the parameters were consistent. Traffic flow data for each approach of the intersection during a pass cycle, as shown in Table 8.

(2) Calculating the optimization objective value

Bring the above parameters into the calculation formula under each objective and the results are shown in Table 9.

As can be seen from Table 8, the per capita delay is equal to the value of the traffic volume index, and the order of passage determined by the travel time is: 4 → 3 → 2 → 1.

**Table 8. Parameters for each approach of the intersection.**

| | Average speed/km·h⁻¹ | Queuing pattern | Automated vehicle penetration rate/% | Number of cars/veh | Number of buses/veh | Left-turn rate/% |
|---|---|---|---|---|---|---|
| 1 | 40 | 2 | 0.125 | 8 | 0 | 50 |
| 2 | 40 | 2 | 0.375 | 8 | 0 | 50 |
| 3 | 40 | 2 | 0.626 | 8 | 0 | 50 |
| 4 | 40 | 2 | 0.875 | 8 | 0 | 50 |

**Table 9. Index value of each approach at intersections.**

| | Per capita delays (—d·s⁻¹) | Travel time (t·s⁻¹) | Traffic volume (veh.pcu⁻¹) |
|---|---|---|---|
| 1 | 0.88 | 14.73 | 8 |
| 2 | 0.88 | 14.47 | 8 |
| 3 | 0.88 | 14.21 | 8 |
| 4 | 0.88 | 13.95 | 8 |

**(3) Determining the optimal right-of-way**

According to each of the three indices corresponds to the optimal order of passage to the intersection of the approach traffic flow from high to low respectively assigned as the I/O indices of the traffic flow in each direction, due to the intersection of the per capita delay in each direction with the traffic volume indicators are equal to the results of the calculations, the results of its assignment are 1 to participate in the subsequent determination of the weights. Through Equations 3–16 and 3–17, the weights of the I/O indices for each direction of traffic flow are obtained, and the results are shown in Table 10.

Based on the I/O indices and their weights, along with Equations 3–20, 3–21, 3–22, and 3–24, the cross-efficiency matrix is obtained. Calculate the mean value of the elements of each row of the matrix as the comprehensive efficiency of each approach lane of the intersection, and order the comprehensive efficiency from highest to lowest to obtain the order of traffic flow of each approach lane. The efficiency value of the west approach is calculated to be 0.49, the efficiency value of the north approach to be 0.59, the efficiency value of the east approach to be 0.73, and the efficiency value of the south approach to be 0.86, and thus the final order of passage is 4→3→2→1, i.e., south approach →east approach →north approach →west approach.

In the above example, 5 parameter values are fixed and 1 parameter value is changed each time, and 1 kind of traffic order can be obtained directly in the second step of the optimization of the objective value calculation, which can be substituted into the third step of the determination of the optimal traffic order, and the optimal traffic order obtained is consistent with the traffic order in the second step. The results show that it is feasible to use the model in this chapter to determine the intersection passing order.

## 4 Experiments and simulations

In order to investigate the regularity and analyze the effectiveness, this section sets up five groups of experiments to investigate the regularity by changing two of the six parameters, and simulate and compare the passing order obtained from the optimization model and the passing order obtained from the actuated control of SUMO software in each experiment, so as to verify the effectiveness of the model.

### 4.1 Simulation platform and scenarios

**4.1.1 Simulation platform.** SUMO is a widely used transportation system simulation software that provides microscopic, partial and multimodal simulation capabilities. In addition, the software is able to simulate the micro-control of

**Table 10. Indices of traffic flow in each direction and the index weights.**

| | Per capita delays | Travel time | Traffic volume |
|---|---|---|---|
| 1 | 1(0.23) | 1(0.43) | 1(0.34) |
| 2 | 1(0.25) | 2(0.40) | 1(0.35) |
| 3 | 1(0.22) | 3(0.46) | 1(0.32) |
| 4 | 1(0.23) | 4(0.46) | 1(0.31) |

traffic flow and the detailed operation of individual vehicles, planning specific trajectories for each vehicle on the road. By writing Python scripts, vehicle control can be translated into executable code that interacts with SUMO through the TraCI interface, enabling precise control of vehicle behavior in SUMO. In the joint simulation platform of SUMO and Python, the Python program is equivalent to the intersection roadside unit in the real traffic scenario, which is responsible for receiving the traffic flow information from SUMO, calculating the optimal vehicle driving commands according to the preset control strategy, and sending these commands back to SUMO through the TraCI interface to guide the actual operation of the vehicle.

### 4.1.2 Scene construction.

(1) Basic parameters of the intersection

In this section, the design of simulation intersections is carried out according to the standards of "*Urban Road Intersection Planning Code*" (GB 50647–2011), with reference to the characteristics of Class A intersections of planar intersections. A four-approach bidirectional four-lane intersection is selected as the simulation model, the length of each road is set to 200 meters, and the lane width is set to 3.5 meters.

(2) Parameters of the Following Model

SUMO simulation, human-driven vehicles, the IDM is selected as the following model, which has many application cases and is more mature in development, and the specific parameter values of this model are shown in Table 11:

In simulation studies of autonomous vehicles, the Cooperative Adaptive Cruise Control (CACC) model, as an advanced car-following strategy, is widely used in autonomous driving scenarios. CACC allows vehicles to exchange real-time

**Table 11. Parameters of the IDM.**

| Parameters | Description | Value |
| --- | --- | --- |
| minGap | Minimum distance between vehicle stop and vehicle in front (m) | 1 |
| accel | Acceleration (m·s$^{-2}$) | 2 |
| decel | Deceleration (m·s$^{-2}$) | 2 |
| emergencyDecel | Maximum deceleration in emergency situations (m·s$^{-2}$) | 9 |
| startupDelay | Additional delay time before starting to drive after stopping | 0 |
| tau | Desired headway (s) | 1 |
| delta | Acceleration impact factor | 4 |

**Table 12. Parameters of the CACC.**

| Parameters | Description | Value |
| --- | --- | --- |
| **minGap** | Minimum distance between vehicle stop and vehicle in front (m) | 0.5 |
| **accel** | Acceleration (m·s$^{-2}$) | 4 |
| **decel** | Deceleration (m·s$^{-2}$) | 4 |
| **emergencyDecel** | Maximum deceleration in emergency situations (m·s$^{-2}$) | 9 |
| **startupDelay** | Additional delay time before starting to drive after stopping | 0 |
| **tau** | Desired headway (s) | 1 |
| **speedControlGainCACC** | Speed deviation factor (speed control mode) | -0.4 |
| **gapClosingControlGainGap** | Positioning deviation factor (gap closing control mode) | 0.005 |
| **gapControlGainGap** | Positioning deviation factor (gap control mode) | 0.45 |
| **collisionAvoidanceGainGap** | Positioning deviation factor (anti-collision mode) | 0.45 |

information such as speed and position through Vehicle-to-Vehicle (V2V) and Vehicle-to-Infrastructure (V2I) communication. This information exchange reduces vehicle reaction times and enhances the stability of traffic flow [32] (Table 12).

## 4.2 Simulation and result analysis

### 4.2.1 Analysis of the results of the optimization of the order of passage.
Five sets of experiments were conducted to investigate the regularity by varying two parameters. Specific parameter settings and comprehensive efficiency are shown in Table 13.

The final order of experiment 1 is 4→3→1→2, i.e., south approach →east approach →west approach → north approach.

The final order of experiment 2 is 4→3→2→1, i.e., south approach →east approach →north approach →west approach.

The final order of experiment 3 is 4→3→2→1, i.e., south approach →east approach →north approach →west approach.

The final order of experiment 4 is 4→3→2→1, i.e., south approach →north approach →west approach →east approach.

The final order of experiment 5 is 1→2→3→4, i.e., west approach →north approach →east approach →south approach.

The above experimental results indicate that

Traffic flows with a higher proportion of buses are given priority over traffic flows with a higher vehicle volume.

Traffic flows with a higher proportion of buses are given priority over traffic flows with a higher proportion of left-turning vehicles.

Table 13. Parameters of each approach and comprehensive efficiency.

| Experiment | Approach | Average speed/km·h⁻¹ | Queuing pattern | Automated vehicle penetration rate/% | Number of cars/veh | Number of buses/veh | Left-turn rate/% | Comprehensive efficiency |
|---|---|---|---|---|---|---|---|---|
| 1 | 1 | 40 | 2 | 0.5 | 9 | 1 | 50 | 0.79 |
|  | 2 | 40 | 2 | 0.5 | 8 | 2 | 50 | 0.66 |
|  | 3 | 40 | 2 | 0.5 | 5 | 3 | 50 | 1.48 |
|  | 4 | 40 | 2 | 0.5 | 4 | 4 | 50 | 1.87 |
| 2 | 1 | 40 | 2 | 0.5 | 8 | 1 | 22 | 0.15 |
|  | 2 | 40 | 2 | 0.5 | 8 | 2 | 40 | 0.28 |
|  | 3 | 40 | 2 | 0.5 | 8 | 3 | 64 | 0.39 |
|  | 4 | 40 | 2 | 0.5 | 8 | 4 | 83 | 0.49 |
| 3 | 1 | 50 | 2 | 0.5 | 8 | 1 | 50 | 0.10 |
|  | 2 | 45 | 2 | 0.5 | 8 | 2 | 50 | 0.29 |
|  | 3 | 40 | 2 | 0.5 | 8 | 3 | 50 | 0.36 |
|  | 4 | 35 | 2 | 0.5 | 8 | 4 | 50 | 0.50 |
| 4 | 1 | 50 | 2 | 0.125 | 8 | 0 | 50 | 0.32 |
|  | 2 | 45 | 2 | 0.375 | 8 | 0 | 50 | 0.36 |
|  | 3 | 40 | 2 | 0.625 | 8 | 0 | 50 | 0.42 |
|  | 4 | 35 | 2 | 0.875 | 8 | 0 | 50 | 0.44 |
| 5 | 1 | 50 | 2 | 0.5 | 8 | 0 | 87.5 | 0.47 |
|  | 2 | 45 | 2 | 0.5 | 8 | 0 | 62.5 | 0.42 |
|  | 3 | 40 | 2 | 0.5 | 8 | 0 | 37.5 | 0.38 |
|  | 4 | 35 | 2 | 0.5 | 8 | 0 | 12.5 | 0.34 |

Traffic flows with a higher proportion of buses are given priority over traffic flows with higher average speed.

Traffic flows with a higher proportion of automated vehicles are given priority over traffic flows with higher average speed.

Traffic flows with higher average speed are given priority over traffic flows with a higher proportion of left-turning vehicles.

**4.2.2 Simulation results analysis.** Using the SUMO simulation software, the passage order calculated from five experiments is compared with the passage order derived from SUMO's built-in actuated signal control and the First-Come, First-Served (FCFS) strategy. In SUMO, actuated signal control dynamically adjusts traffic signal phases by continuously monitoring vehicle flow in real-time. This mechanism relies on vehicle detectors installed at the intersection, which collect flow data from each lane and transmit it to the signal control system. Based on this data, the system determines whether to extend or shorten the green light duration for specific directions, thereby optimizing intersection efficiency. The First-Come, First-Served (FCFS) strategy refers to prioritizing vehicles that arrive first at an intersection or traffic facility, allowing them to pass through before others. Each experiment conducts a simulation based on the traffic conditions in Table 13 using the FCFS strategy, actuated signal control, and optimal control, respectively. In each simulation, each approach passes a single flow of traffic, and during the simulation using optimal control, the passing order remains unchanged. Each passage cycle is defined as the complete passage of vehicles from all four directions through the intersection. The per capita delays and travel time are obtained for comparative analysis. The results are shown in Figs 5 and 6.

As indicated by Figs 5 and 6, the optimized order of passage results in significantly reduced per capita delay and travel time compared to actuated control and FCFS. Therefore, the optimization method proposed in this study effectively enhances intersection traffic efficiency.

## 5 Conclusions

In this paper, we have researched the intersection passing strategies in the mixed environment of automated and manual driving by using Data Envelopment Analysis (DEA). A method to optimize the intersection passing strategies is proposed by selecting the influencing factors such as average speed, number of cars, penetration of automated vehicles, queuing

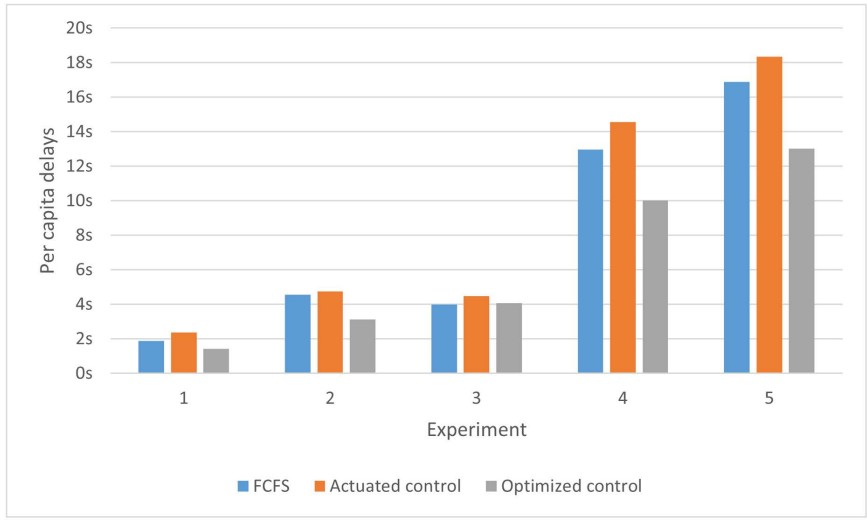

**Fig 5. Per capita delay comparison of drive control, FCFS and optimal control.**

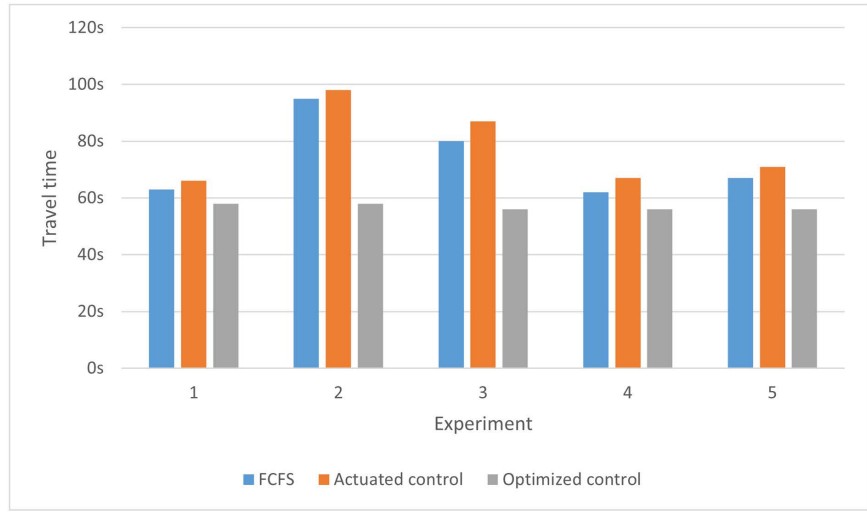

**Fig 6. Travel time comparison of drive control, FCFS and optimal control.**

pattern, left-turn rate, and number of buses, and considering the per capital delay, travel time and traffic volume as the optimization objectives. Simulation results indicate that, compared to traditional actuated control methods and FCFS strategy at intersections, this strategy significantly reduces the per capita delay and travel time. In mixed traffic environments, the DEA-based optimization method proposed in this research is of significant importance for enhancing intersection efficiency. The key conclusions and findings of this study are as follows:

Advantages of the Model: The DEA method enables the comprehensive consideration of multiple influencing factors (such as per capita delay, travel time, and traffic volume) and flexibly adapts to different traffic flow configurations. Compared to traditional single-objective optimization methods, this approach demonstrates significant advantages in enhancing intersection efficiency.

Improvements to Traffic Operations: By prioritizing the passage order for buses, followed by autonomous vehicles and vehicles with higher average speeds, this method effectively reduces per capita delay and travel time, contributing to improved overall intersection efficiency. For practical applications, the optimization model proposed in this study can serve as a reference strategy, providing managers with a more flexible tool for optimizing intersection passage order.

This study primarily focused on per capita delay, travel time, and traffic volume. Future research could consider additional optimization objectives, such as vehicle emissions, fuel consumption, and safety, to achieve more comprehensive optimization. This research primarily addresses single-intersection control and does not thoroughly consider the optimization of benefits at consecutive intersections. Future research could focus on a more detailed and comprehensive analysis of consecutive intersections. Additionally, the model assumes that vehicles strictly follow their lanes without considering potential lane-changing or yielding behaviors that may occur in real-world scenarios. Future research could incorporate lane-changing behaviors to enhance the model's realism and applicability. This study only determined the optimal traffic order of the vehicles in each direction under different traffic demands, but did not conduct specific research on signal timing of the signal lights used to guide the vehicles through the intersection in order. Future studies can further study signal timing based on this study.

## Author contributions

**Conceptualization:** Jiajun Shen, Guanyu Fu.

**Formal analysis:** Yu Wang.

**Funding acquisition:** Jiajun Shen.

**Investigation:** Yu Wang.

**Methodology:** Jiajun Shen, Zhipeng Zhou, Guanyu Fu.

**Resources:** Jiajun Shen.

**Software:** Zhipeng Zhou.

**Supervision:** Jiajun Shen.

**Validation:** Jiajun Shen, Zhipeng Zhou, Yu Wang.

**Writing – original draft:** Zhipeng Zhou.

**Writing – review & editing:** Jiajun Shen, Zhipeng Zhou.

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
