## [Decision Letter · Decision Letter 0]

24 Jan 2025

PONE-D-24-52478Intersection passing strategies for human-driven and autonomous vehicles in mixed traffic using DEAPLOS ONE

Dear Dr. zhou,

Thank you for submitting your manuscript to PLOS ONE. After careful consideration, we feel that it has merit but does not fully meet PLOS ONE’s publication criteria as it currently stands. Therefore, we invite you to submit a revised version of the manuscript that addresses the points raised during the review process.

See comments below. 

We look forward to receiving your revised manuscript.

Kind regards,

MJ Booysen

Academic Editor

PLOS ONE

Journal requirements: When submitting your revision, we need you to address these additional requirements. 1. Please ensure that your manuscript meets PLOS ONE's style requirements, including those for file naming. The PLOS ONE style templates can be found at https://journals.plos.org/plosone/s/file?id=wjVg/PLOSOne_formatting_sample_main_body.pdf and https://journals.plos.org/plosone/s/file?id=ba62/PLOSOne_formatting_sample_title_authors_affiliations.pdf. 2. Please amend your list of authors on the manuscript to ensure that each author is linked to an affiliation. Authors’ affiliations should reflect the institution where the work was done (if authors moved subsequently, you can also list the new affiliation stating “current affiliation:….” as necessary). 3. Please note that PLOS ONE has specific guidelines on code sharing for submissions in which author-generated code underpins the findings in the manuscript. In these cases, we expect all author-generated code to be made available without restrictions upon publication of the work. Please review our guidelines at https://journals.plos.org/plosone/s/materials-and-software-sharing#loc-sharing-code and ensure that your code is shared in a way that follows best practice and facilitates reproducibility and reuse. 4. Thank you for stating the following financial disclosure:  [This research was funded by the Humanities and Social Sciences Research Planning Fund from the Ministry of Education of China, Grant No. 23YJAZH122. It was conducted as part of the project titled "Safety Assessment and Efficiency Optimization of Road Intersections in Mixed Traffic Environments of Manual and Autonomous Driving."].  Please state what role the funders took in the study.  If the funders had no role, please state: ""The funders had no role in study design, data collection and analysis, decision to publish, or preparation of the manuscript."" If this statement is not correct you must amend it as needed. Please include this amended Role of Funder statement in your cover letter; we will change the online submission form on your behalf. 5. Thank you for stating the following in the Acknowledgments Section of your manuscript: [This research was funded by the Humanities and Social Sciences Research Planning Fund from the Ministry of Education of China, Grant No. 23YJAZH122. It was conducted as part of the project titled "Safety Assessment and Efficiency Optimization of Road Intersections in Mixed Traffic Environments of Manual and Autonomous Driving." The views and conclusions expressed in this paper are solely those of the authors.]We note that you have provided funding information that is not currently declared in your Funding Statement. However, funding information should not appear in the Acknowledgments section or other areas of your manuscript. We will only publish funding information present in the Funding Statement section of the online submission form. Please remove any funding-related text from the manuscript and let us know how you would like to update your Funding Statement. Currently, your Funding Statement reads as follows:  [This research was funded by the Humanities and Social Sciences Research Planning Fund from the Ministry of Education of China, Grant No. 23YJAZH122. It was conducted as part of the project titled "Safety Assessment and Efficiency Optimization of Road Intersections in Mixed Traffic Environments of Manual and Autonomous Driving."] Please include your amended statements within your cover letter; we will change the online submission form on your behalf. 6. We note that your Data Availability Statement is currently as follows: [All relevant data are within the manuscript and its Supporting Information files.] Please confirm at this time whether or not your submission contains all raw data required to replicate the results of your study. Authors must share the “minimal data set” for their submission. PLOS defines the minimal data set to consist of the data required to replicate all study findings reported in the article, as well as related metadata and methods (https://journals.plos.org/plosone/s/data-availability#loc-minimal-data-set-definition). For example, authors should submit the following data: - The values behind the means, standard deviations and other measures reported;- The values used to build graphs;- The points extracted from images for analysis. Authors do not need to submit their entire data set if only a portion of the data was used in the reported study. If your submission does not contain these data, please either upload them as Supporting Information files or deposit them to a stable, public repository and provide us with the relevant URLs, DOIs, or accession numbers. For a list of recommended repositories, please see https://journals.plos.org/plosone/s/recommended-repositories. If there are ethical or legal restrictions on sharing a de-identified data set, please explain them in detail (e.g., data contain potentially sensitive information, data are owned by a third-party organization, etc.) and who has imposed them (e.g., an ethics committee). Please also provide contact information for a data access committee, ethics committee, or other institutional body to which data requests may be sent. If data are owned by a third party, please indicate how others may request data access.

Additional Editor Comments:

Thank you for your submission.

The reviewers' feedback has been received, and both recommend a Major Revision. Please carefully address all the feedback, and pay specific attention to the following items pointed out by the reviewers:

- Justify model assumptions and parameters. The theoretical projection of the real world seems to be substantially simplified.

- Compare with existing literature, both in terms of approach/method and results. Also check bold statements made about gaps in current literature.

- Highlight limitations of the presented work.

I want to urge the authors to submit a substantially reworked version, which carefully and wholly addresses all the comments by the reviewers, since they do raise important and substantial concerns.

Reviewers' comments:

Reviewer's Responses to Questions

**Comments to the Author**

1. Is the manuscript technically sound, and do the data support the conclusions?

Reviewer #1: Yes

Reviewer #2: Partly

2. Has the statistical analysis been performed appropriately and rigorously? 

Reviewer #1: N/A

Reviewer #2: N/A

3. Have the authors made all data underlying the findings in their manuscript fully available?

Reviewer #1: No

Reviewer #2: No

4. Is the manuscript presented in an intelligible fashion and written in standard English?

Reviewer #1: Yes

Reviewer #2: Yes

5. Review Comments to the Author

Reviewer #1: In the present work, the authors proposed a right-of-way optimization model considering multi-objective DEA evaluation for intersections in mixed driving environments with automated and human driving. The construction of the model and derivation of the formula are very detailed. However, the paper also suffers some limits. My comments are as follows:

1. If it is not required by the journal, it is recommended to mark the number of parts of the article with the serial number before each chapter. Otherwise, it will appear that the structure of the article is somewhat chaotic.

2. How are some indicators, such as Queuing pattern, specifically defined? The article seems to have omitted this information.

3. The manuscript assumes all vehicles run along traffic lanes. However, the vehicular movement inside intersections is two-dimensional. This assumption should be clarified. The relevant existing studies should be cited. E.g., Microscopic traffic modeling inside intersections Interactions between drivers, Transportation Science, 2023; Unprotected left-turn behavior model capturing path variations at intersections, IEEE-T-ITS, 2023.

4. The authors like to use large paragraphs to describe a certain problem, but the logical structure of the language expression is not clear enough. For example, in the section on influencing factors, the authors use a long paragraph to explain the various factors that need to be considered when formulating traffic strategies, but the overall structure is somewhat loose and lacks a clear logical thread. It is recommended that the authors split this paragraph into several sections, each focusing on one or several related factors to make the content clearer and easier to understand.

5. Only one comparison is used, which doesn't seem convincing enough. More experiments are suggested.

Reviewer #2: The authors analyze strategies for intersections passing when both autonomous and human-driven vehicles are present.

The literature review is incomplete about mixed traffic control. Recent works have been dedicated to the optimization and analysis of traffic intersections under mixed traffic scenarios and have not been referenced. You state that “…researches involving the passage order of intersections in mixed traffic environments consider a single optimization objective and are unable to weigh the …”. This is not quite true. See for example,

Changxin Wan, Xiaonian Shan, Peng Hao, Guoyuan Wu, Multi-objective coordinated control strategy for mixed traffic with partially connected and automated vehicles in urban corridors, Physica A: Statistical Mechanics and its Applications, Volume 635, 2024.

You should clearly identify the limitations of previous and recent works about the same subject and how do you expect to overcome some of their potential limitations.

How do you control the order of passage? The intersections do not have traffic signals. So, how do you control the movement of human-driven vehicles?

Why is the travel time reducing as the penetration rate increase?

In tables 11 and 12, what is the minimum distance between human-driven and autonomous vehicles?

The model defines the passing order of vehicles based on intersection scenarios. Are you considering the same conditions for all directions? It is not clear how you determine the order of passage from the traffic scenario. Is the order fixed during the complete simulation? When do the system decide to switch to a different traffic phase?

What do you consider to be the traditional actuated control mechanisms?

How do you compare with other optimization strategies?

How do you compare against intelligent solutions based, for example, on reinforcement learning?

6. PLOS authors have the option to publish the peer review history of their article (what does this mean? ). If published, this will include your full peer review and any attached files.

**Do you want your identity to be public for this peer review?** For information about this choice, including consent withdrawal, please see our Privacy Policy .

Reviewer #1: No

Reviewer #2: No

---

## [Author Response · Author response to Decision Letter 1]

7 Feb 2025

Response to Reviewers

Manuscript number: PONE-D-24-52478

Title: Intersection passing strategies for human-driven and autonomous vehicles in mixed traffic using DEA

Dear editor and reviewers,

We are very grateful for your constructive comments and suggestions for our manuscript entitled “Intersection passing strategies for human-driven and autonomous vehicles in mixed traffic using DEA” (ID: PONE-D-24-52478). As you are concerned, there are several problems that need to be addressed. According to your nice suggestions, we have made extensive corrections to our previous manuscript, the detailed corrections are listed below.

We sincerely thank the editor and all reviewers for their valuable feedback that we have used to improve the quality of our manuscript. The reviewer comments are laid out below in italicized font and specific concerns have been numbered. Our response is given in normal font and changes/ additions to the manuscript are given in the yellow text.

Respond to the editor’s comments

Q1: Please ensure that your manuscript meets PLOS ONE's style requirements, including those for file naming.

Response 1: Thank you for your reminder. I have carefully revised my manuscript according to the PLOS ONE template to ensure it meets the required formatting standards.

Q2: Please amend your list of authors on the manuscript to ensure that each author is linked to an affiliation. Authors’ affiliations should reflect the institution where the work was done.

Response 2: Thank you for your reminder. I have ensured that each author is linked to their corresponding affiliation.

Q3: Please note that PLOS ONE has specific guidelines on code sharing for submissions in which author-generated code underpins the findings in the manuscript. In these cases, we expect all author-generated code to be made available without restrictions upon publication of the work.

Response 3: Thank you for your valuable feedback. This study does not involve any code, so no code has been uploaded.

Q4: Thank you for stating the following financial disclosure:

[This research was funded by the Humanities and Social Sciences Research Planning Fund from the Ministry of Education of China, Grant No. 23YJAZH122. It was conducted as part of the project titled "Safety Assessment and Efficiency Optimization of Road Intersections in Mixed Traffic Environments of Manual and Autonomous Driving."].

Response 4: Apologies for the omission. The funder provides financial support for the research project. I have updated the funder role statement in the cover letter.

Q5: Thank you for stating the following in the Acknowledgments Section of your manuscript:

[This research was funded by the Humanities and Social Sciences Research Planning Fund from the Ministry of Education of China, Grant No. 23YJAZH122. It was conducted as part of the project titled "Safety Assessment and Efficiency Optimization of Road Intersections in Mixed Traffic Environments of Manual and Autonomous Driving." The views and conclusions expressed in this paper are solely those of the authors.]

[This research was funded by the Humanities and Social Sciences Research Planning Fund from the Ministry of Education of China, Grant No. 23YJAZH122. It was conducted as part of the project titled "Safety Assessment and Efficiency Optimization of Road Intersections in Mixed Traffic Environments of Manual and Autonomous Driving."]

Response 5: Apologies for this issue. I have removed the funding statement from the manuscript and included it in the cover letter.

Q6: We note that your Data Availability Statement is currently as follows: [All relevant data are within the manuscript and its Supporting Information files.]

Please confirm at this time whether or not your submission contains all raw data required to replicate the results of your study. Authors must share the “minimal data set” for their submission. PLOS defines the minimal data set to consist of the data required to replicate all study findings reported in the article, as well as related metadata and methods

Response 6: Thank you for your valuable suggestion. The data used in this study are assumed values for explaining the research content and verifying the feasibility of the proposed method. These data have been fully presented in Tables 5, 8, and 13 of the manuscript.

Respond to additional editor comments:

Q1: Justify model assumptions and parameters. The theoretical projection of the real world seems to be substantially simplified.

Response 1: Thank you for pointing this out. To justify the assumptions, I have cited references 29 and 30 in lines 202 to 205 of the manuscript to explain their validity.

Q2: Compare with existing literature, both in terms of approach/method and results. Also check bold statements made about gaps in current literature.

Response 2: Apologies for this issue. I have revised the literature review section to refine statements about existing studies and have cited references 26, 27, and 28 to ensure a more comprehensive comparison. Additionally, I have restated the contributions of this study and its advantages over existing research in lines 180 to 197 of the manuscript.

Q3: Highlight limitations of the presented work.

Response 3: Thank you for your suggestion. I have added a discussion on the limitations of this study in the "Conclusion" section, specifically in lines 629 to 637, to ensure the objectivity and completeness of the research findings. 

Respond to Reviewer #1’ comments

Q1: If it is not required by the journal, it is recommended to mark the number of parts of the article with the serial number before each chapter. Otherwise, it will appear that the structure of the article is somewhat chaotic.

Response 1: Thank you for your insightful suggestion. I have added numbering to the sections of the paper to improve its structure.

Q2: How are some indicators, such as Queuing pattern, specifically defined? The article seems to have omitted this information.

Response 2: Thank you for pointing out this problem. I have now clarified the definition of the indicators in Section 3.1 of the article. Specifically, from lines 232 to 239, the queuing pattern is explained as the specific queuing order of the two types of vehicles in a mixed traffic flow containing CAVs (Connected Autonomous Vehicles) and HVs (Human-driven Vehicles). The queuing patterns can be categorized into three types: the first type, where all CAVs are positioned at the front of the queue; the second type, where CAVs and HVs are randomly mixed within the queue; and the third type, where all HVs are positioned at the front of the queue.

Q3: The manuscript assumes all vehicles run along traffic lanes. However, the vehicular movement inside intersections is two-dimensional. This assumption should be clarified. The relevant existing studies should be cited. E.g., Microscopic traffic modeling inside intersections Interactions between drivers, Transportation Science, 2023; Unprotected left-turn behavior model capturing path variations at intersections, IEEE-T-ITS, 2023.

Response 3: Thank you for your valuable suggestion. I have addressed this point in the manuscript by citing two relevant studies in lines 202 to 205 to explain the rationale behind assuming that all vehicles run along traffic lanes.

Q4: The authors like to use large paragraphs to describe a certain problem, but the logical structure of the language expression is not clear enough. For example, in the section on influencing factors, the authors use a long paragraph to explain the various factors that need to be considered when formulating traffic strategies, but the overall structure is somewhat loose and lacks a clear logical thread. It is recommended that the authors split this paragraph into several sections, each focusing on one or several related factors to make the content clearer and easier to understand.

Response 4: Thank you for your thoughtful feedback. I have revised the manuscript by splitting the paragraph into several sections in Section 3.1, from lines 225 to 242, each focusing on one or more related factors. Your suggestion has greatly improved the clarity and readability of the manuscript.

Q5: Only one comparison is used, which doesn't seem convincing enough. More experiments are suggested.

Response 5: We are grateful for your constructive suggestions. I have added a comparison experiment using the First-Come, First-Served (FCFS) strategy in Section 4.2.2 and updated Figures 5 and 6 to make the research results more convincing. The results indicate that the optimized order of passage significantly reduces per capita delay and travel time compared to both actuated control and FCFS.

Respond to Reviewer #2’ comments

Q1: The literature review is incomplete about mixed traffic control. Recent works have been dedicated to the optimization and analysis of traffic intersections under mixed traffic scenarios and have not been referenced. You state that “…researches involving the passage order of intersections in mixed traffic environments consider a single optimization objective and are unable to weigh the …”. This is not quite true. See for example,

Changxin Wan, Xiaonian Shan, Peng Hao, Guoyuan Wu, Multi-objective coordinated control strategy for mixed traffic with partially connected and automated vehicles in urban corridors, Physica A: Statistical Mechanics and its Applications, Volume 635, 2024.

You should clearly identify the limitations of previous and recent works about the same subject and how do you expect to overcome some of their potential limitations.

Response 1: Thank you very much for pointing out this mistake. I have added relevant literature on intersection passage strategies in mixed traffic environments in Section 2, from lines 166 to 177 (references 26, 27, and 28). Additionally, in lines 180 to 197, I have explained how we overcome the limitations of these studies.

Q2: How do you control the order of passage? The intersections do not have traffic signals. So, how do you control the movement of human-driven vehicles?

Response 2: Apologies for the oversight in the manuscript. The intersections in this study are signalized intersections, and we control the order of passage by switching the signal phases to determine the sequence in which vehicles from different approaches pass through the intersection. I have added this clarification in lines 210 to 211 of the manuscript.

Q3: Why is the travel time reducing as the penetration rate increase?

Response 3: Thank you for raising this question. CAVs (Connected Autonomous Vehicles) have shorter reaction times and can communicate with each other, which enhances coordination between vehicles and reduces travel time. AVs are able to maintain optimal speeds and spacing, minimizing delays caused by unstable driving behaviors. Therefore, as the penetration rate of autonomous vehicles increases, travel time will decrease. I have added an explanation regarding this in lines 334 to 338 of the manuscript.

Q4: In tables 11 and 12, what is the minimum distance between human-driven and autonomous vehicles?

Response 4: Thank you for raising this question, and I apologize for not providing a clear explanation of the definition of ‘minGap’. ‘minGap’ refers to the minimum distance between a vehicle and the vehicle in front when the vehicle is at a stop. This definition has been updated in Tables 11 and 12. Therefore, when a CAV follows an HV, the minimum distance between the two vehicles is 0.5 meters, and when an HV follows a CAV, the minimum distance is 1 meter.

Q5: The model defines the passing order of vehicles based on intersection scenarios. Are you considering the same conditions for all directions? It is not clear how you determine the order of passage from the traffic scenario. Is the order fixed during the complete simulation? When do the system decide to switch to a different traffic phase?

Response 5: Thank you for raising these valuable questions. Based on the data from the five experiments in Table 13, there are differences in Average speed, Queuing pattern, Automated vehicle penetration, Number of cars, Number of buses, and Left-turn rate for all directions. The passing order is determined by comparing the comprehensive efficiency of each direction, calculated using Data Envelopment Analysis (DEA), with the direction having higher efficiency prioritized for passage. Since each simulation cycle corresponds to one platoon passing through each approach and only one cycle is performed in each simulation, the order of passage remains fixed throughout the simulation. I have added this clarification in lines 596 to 597 of the manuscript. Regarding the switching to different traffic phases, vehicles with a headway of less than 3 seconds are grouped into a single platoon, and each signal cycle allows one platoon to pass through each approach. The passing order is recalculated for each cycle. I have added this explanation in lines 211 to 213 of the manuscript.

Q6: What do you consider to be the traditional actuated control mechanisms?

Response 6: Thank you for raising this important question. In SUMO, actuated signal control dynamically adjusts traffic signal phases by continuously monitoring vehicle flow in real-time. This mechanism relies on vehicle detectors installed at the intersection, which collect flow data from each lane and transmit it to the signal control system. Based on this data, the system determines whether to extend or shorten the green light duration for specific directions, thereby optimizing intersection efficiency. To make the content clearer, I have added this explanation in lines 588 to 592 of the manuscript.

Q7: How do you compare with other optimization strategies?

Response 7: I appreciate your interest in this matter. In comparison with other optimization strategies, the contribution of this study lies in the proposed optimization model, which comprehensively considers various factors, including average speed, number of cars, penetration of automated vehicles, queuing pattern, left-turn rate, and number of buses. This model provides a more holistic representation of the complex traffic flow characteristics at intersections. Additionally, the optimization objectives focus on improving intersection passage efficiency, incorporating multi-dimensional goals such as per capita delay, travel time, and traffic volume, thereby making the optimization results more comprehensive and accurate. These influencing factors and optimization objectives provide a more precise and scientifically grounded basis for optimizing intersection passage efficiency. Regarding the advantages over other optimization strategies, I have added supplementary information in lines 187 to 197 of the manuscript.

Q8: How do you compare against intelligent solutions based, for example, on reinforcement learning?

Response 8: Thank you for your insightful question. Undoubtedly, intelligent solutions, especially those based on reinforcement learning, have become quite popular in the field of intelligent transportation. However, methods based on reinforcement learning rely on large-scale data training, and the complexity of traffic environments may make it difficult to accurately map the states, actions, and rewards in reinforcement learning, potentially resulting in poor training outcomes. In contrast, the method proposed in this study effectively handles complex traffic environments without the need for large-scale

---

## [Decision Letter · Decision Letter 1]

18 Feb 2025

PONE-D-24-52478R1Intersection passing strategies for human-driven and autonomous vehicles in mixed traffic using DEAPLOS ONE

Dear Dr. zhou,

Thank you for submitting your manuscript to PLOS ONE. After careful consideration, we feel that it has merit but does not fully meet PLOS ONE’s publication criteria as it currently stands. Therefore, we invite you to submit a revised version of the manuscript that addresses the points raised during the review process.

Please see the comments summarised below. 

We look forward to receiving your revised manuscript.

Kind regards,

MJ Booysen

Academic Editor

PLOS ONE

Additional Editor Comments:

Please see the valid comments by Reviewer 2:

- Code of the proposed solution should be made available. At least to the reviewer, and preferably to the public, to ensure reproducibility.

- Relationship between the traffic lights and the metrics.

- Comparison with existing approaches required.

- Graphs must have units

Reviewers' comments:

Reviewer's Responses to Questions

**Comments to the Author**

1. If the authors have adequately addressed your comments raised in a previous round of review and you feel that this manuscript is now acceptable for publication, you may indicate that here to bypass the “Comments to the Author” section, enter your conflict of interest statement in the “Confidential to Editor” section, and submit your "Accept" recommendation.

Reviewer #1: All comments have been addressed

Reviewer #2: (No Response)

2. Is the manuscript technically sound, and do the data support the conclusions?

Reviewer #1: Yes

Reviewer #2: Partly

3. Has the statistical analysis been performed appropriately and rigorously? 

Reviewer #1: Yes

Reviewer #2: N/A

4. Have the authors made all data underlying the findings in their manuscript fully available?

Reviewer #1: Yes

Reviewer #2: No

5. Is the manuscript presented in an intelligible fashion and written in standard English?

Reviewer #1: Yes

Reviewer #2: Yes

6. Review Comments to the Author

Reviewer #1: The authors have adequately addressed your comments raised in a previous round of review. I have no more comments.

Reviewer #2: To ensure reproducibility of the results, the code of the proposed solution should be made public on a website.

The dynamics of traffic lights should be included, that is, the green, red and yellow times and how they affect the metrics.

Complex or not, approaches based on DRL exist with good results. The work should be compared with these approaches.

The units are missing in the graphs. For example, in Figures 5 and 6, what are the time units?

7. PLOS authors have the option to publish the peer review history of their article (what does this mean? ). If published, this will include your full peer review and any attached files.

**Do you want your identity to be public for this peer review?** For information about this choice, including consent withdrawal, please see our Privacy Policy .

Reviewer #1: No

Reviewer #2: No

---

## [Author Response · Author response to Decision Letter 2]

2 Mar 2025

Response to Reviewers

Manuscript number: PONE-D-24-52478

Title: Intersection passing strategies for human-driven and autonomous vehicles in mixed traffic using DEA

Dear editor and reviewers,

Thank you for giving us the opportunity to revise and resubmit our manuscript entitled “Intersection passing strategies for human-driven and autonomous vehicles in mixed traffic using DEA” (ID: PONE-D-24-52478). We sincerely appreciate the reviewers' constructive feedback and have carefully addressed all comments. According to your nice suggestions, we have made extensive corrections to our previous manuscript, the detailed corrections are listed below.

We sincerely thank the editor and all reviewers for their valuable feedback that we have used to improve the quality of our manuscript. The reviewer comments are laid out below in italicized font and specific concerns have been numbered. Our response is given in normal font and changes/ additions to the manuscript are given in the yellow text.

Respond to additional editor comments:

Q1: Code of the proposed solution should be made available. At least to the reviewer, and preferably to the public, to ensure reproducibility.

Response 1: We appreciate the comment regarding the availability of code for reproducibility. In our study, all results are derived directly from the formulas in Section 3.3 (Formulas 3-16 to 3-24) of the paper. These formulas do not require computational algorithms to obtain the results, and thus, no code is necessary for the study. Therefore, we are unable to provide code for this research.

Q2: Relationship between the traffic lights and the metrics.

Response 2: We acknowledge the importance of the dynamics of traffic lights in traffic flow analysis. However, our study primarily focuses on determining the optimal intersection passage order under varying traffic demands, and as such, traffic signal timing was not considered within the scope of this work. We recognize the significant impact of traffic signal timing on overall intersection efficiency, and this could be a direction for future research. We have added a discussion of this limitation on lines 638 to 641 in the manuscript, and we hope to explore this aspect in future studies.

Q3: Comparison with existing approaches required.

Response 3: We thank Reviewer #2 for suggesting a comparison with deep reinforcement learning (DRL) methods. In existing research, many DRL-based approaches have been applied to traffic signal control and intersection optimization, where reinforcement learning agents automatically adjust traffic signals to optimize traffic flow. However, our study proposes an optimization method that directly calculates the intersection passage order based on traffic flow conditions, focusing on determining the optimal passage order. Future research could combine our approach with DRL methods to optimize traffic signals at intersections. 

Respond to Reviewer #2’ comments

Q1: To ensure reproducibility of the results, the code of the proposed solution should be made public on a website.

Response 1: Thank you for raising this important point regarding reproducibility. In our study, all results are derived directly from formulas 3-16 to 3-24 in Section 3.3 of the paper. These formulas do not require computational algorithms to obtain the results. Therefore, no code can be provided for this study.

Q2: The dynamics of traffic lights should be included, that is, the green, red and yellow times and how they affect the metrics.

Response 2: Thank you for the detailed feedback on our research. Regarding the traffic signal timing issue, our study currently focuses on determining the optimal intersection passage order under different traffic demands, and thus, traffic signal timing is not considered within the scope of this study. We recognize that traffic signal timing plays an important role in the overall traffic efficiency of intersections, and it can be considered as a direction for future research. I have added a discussion of this limitation in lines 638 to 641 of the manuscript, and we hope to explore it in future studies.

Q3: Complex or not, approaches based on DRL exist with good results. The work should be compared with these approaches.

Response 3: Thank you for your valuable feedback. It is undeniable that many deep reinforcement learning (DRL)-based methods have been applied to traffic signal control and intersection throughput optimization in existing research. These methods typically rely on large-scale data training, where reinforcement learning agents automatically adjust traffic signal timing to optimize traffic flow and reduce vehicle delays. The focus of these studies is often on long-term decision-making, such as dynamically adjusting the signal cycle and duration, and they tend to prioritize optimizing signal control rather than directly determining the passage order.

However, this study proposes an optimization method that directly calculates the intersection passage order based on traffic flow conditions. This method focuses on determining the optimal passage order using real-time traffic flow data and specific traffic demand.

Although the focus of this study is on directly calculating the intersection passage order based on traffic flow conditions, without involving specific signal timing research, future studies may consider applying DRL methods to optimize signal timing. Combining this with the current passage order optimization could further enhance the overall throughput of intersections.

Q4: The units are missing in the graphs. For example, in Figures 5 and 6, what are the time units?

Response 4: Thank you for pointing out the issue with the missing units in the graphs. Regarding Figures 5 and 6, I have made the necessary revisions and added the appropriate time units to the graphs. Thank you again for your careful review.

We sincerely appreciate the reviewers’ insightful feedback, which has significantly improved our manuscript. We look forward to your response and appreciate your time and consideration.

Your faithfully

On behalf of co-authors (Professor Dr Jiajun Shen; Guanyu Fu; Yu Wang)

Zhipeng Zhou

College of Architectural Science and Engineering, Yangzhou University,

E-mail: zzpash@outlook.com

---

## [Decision Letter · Decision Letter 2]

9 Mar 2025

Intersection passing strategies for human-driven and autonomous vehicles in mixed traffic using DEA

PONE-D-24-52478R2

Dear Dr. zhou,

We’re pleased to inform you that your manuscript has been judged scientifically suitable for publication and will be formally accepted for publication once it meets all outstanding technical requirements.

Kind regards,

MJ Booysen

Academic Editor

PLOS ONE

---

## [Editor Report · Acceptance letter]

PONE-D-24-52478R2

PLOS ONE

Dear Dr. zhou,

I'm pleased to inform you that your manuscript has been deemed suitable for publication in PLOS ONE. Congratulations! Your manuscript is now being handed over to our production team.

Kind regards,

on behalf of

Professor MJ Booysen

Academic Editor

PLOS ONE